# Efficient Recovery of Feldspar, Quartz, and Kaolin from Weathered Granite

Hongjun Huang [1,2,*], Shihan Li [1,2], Haoran Gou [1,2], Ning Zhang [1,2] and Liming Liu [3]

1   School of Minerals Processing and Bioengineering, Central South University, Changsha 410083, China; lshyd97@163.com (S.L.); 215612114@csu.edu.cn (H.G.); 225611052@csu.edu.cn (N.Z.)
2   Engineering Research Center of Ministry of Education for Carbon Emission Reduction in Metal Resource Exploitation and Utilization, Central South University, Changsha 410083, China
3   College of Chemistry, Xiangtan University, Xiangtan 411100, China; 18074678820@163.com
*   Correspondence: 207049@csu.edu.cn

**Abstract:** Weathered granite contains a high concentration of feldspar, quartz, and kaolin. However, while it becomes rich in clay due to strong physical weathering, the granite minerals that are not fully weathered are still very hard, which makes the grinding process more difficult and limits its use. This study proposes a multi-step process involving grinding, desliming, and flotation to address this issue. The study determines the appropriate grinding time and power index for the original ore, as well as the optimal desliming method using a hydrocyclone. To remove iron-containing impurities like mica, a combination of NaOL/BHA/A CO collectors is used for the reverse rough flotation of quartz. Additionally, a combination of DDA/SDS collectors is employed to separate quartz and feldspar through flotation, resulting in a quartz product with a silicon dioxide content of 99.51%. The objective of efficiently recycling feldspar, quartz, and kaolin from weathered granite is accomplished. Additionally, the inclusion of intermediate mineral components as by-products of feldspar and raw materials for aerated bricks is introduced, resulting in the complete utilization of all components. This innovative approach ensures a clean and environmentally friendly process, eliminating the need for solid waste disposal.

**Keywords:** weathered granite; feldspar; quartz; kaolin; recovery





## 1. Introduction

Weathered granite is a layer of soil formed on the parent rock when granite undergoes intense physical weathering and is affected by impure surface water, such as rainwater, river water, seawater, and groundwater containing carbonic acid, causing chemical changes [1–3]. Its main mineral composition includes clay minerals such as feldspar ($KAlSi_3O_8$-$NaAlSi_3O_8$), quartz ($SiO_2$), mica ($KAl_2(AlSi_3O_{10})(OH)_2$), and kaolinite ($Al_4[Si_4O_{10}](OH)_8$) [4,5]. Feldspar and other minerals with poor weathering resistance gradually lose their $K^+$ and $Na^+$ content under the action of water and carbon dioxide, forming the clay mineral kaolinite [6,7]. This creates the characteristics of high clay content and complex mineral composition in weathered granite [8,9]. At the same time, these characteristics also lead to the generation of a large amount of waste, which puts pressure on the environment [10]. Therefore, it is of great significance to conduct experiments and research on the efficient and comprehensive utilization of weathered granite [11].

Due to its unique geographical location, China has a large amount of weathered granite that urgently needs to be comprehensively utilized [12–14]. By using beneficiation techniques, quartz, sodium feldspar, and potassium feldspar can be separated from weathered granite, removing colored impurities such as mica, iron, and titanium minerals [15–17]. The combination of gravity separation and magnetic separation can reduce the content of colored impurities [18]. Adding flotation to the magnetic separation process can further reduce the iron content in the granite, as well as remove mica, iron, and titanium

mineral impurities, resulting in feldspar and quartz concentrates. Flotation can produce concentrates with fewer impurities compared to magnetic separation, and both methods can meet the raw material requirements of the ceramic industry. Among them, feldspar products with an iron impurity content of less than 1% and quartz products with a silica content of more than 99% meet the sales standards [19–21].

Most of the current studies focus on the valuable components, such as quartz, feldspar, and mica, that are separated from weathered granite during the sorting process. There are few studies conducted on the treatment of medium ore, fine-grained slime, and magnetic concentrate with high iron content [22–24]. These materials are often discarded as tailings waste and not fully utilized. This failure to make full use of resources puts greater pressure on the ecological environment.

In order to utilize the characteristics of weathered granite in Guangdong, this experiment utilizes various combined beneficiation methods such as physical separation, flotation, and chemical beneficiation to enrich and separate the components of feldspar, quartz, and kaolin. The process parameters are optimized to determine the best process flow and ultimately achieve satisfactory laboratory results. Next, the primary product is tested for improved whiteness to meet the quality requirements of ceramic raw materials. Performance tests are also conducted to determine the optimal product plan. Finally, a detailed small-scale experiment is conducted to achieve favorable results and determine the best product plan. This research has developed technologically and economically feasible new technologies for the comprehensive utilization of innovative solutions that achieve the efficient, clean, waste-free, full utilization of all products.

## 2. Materials and Methods

### 2.1. Mineralogy and Reagents

In this study, the experimental samples were obtained from Fengsheng Mining Co., Ltd. in Guangdong, China. To ensure the representativeness of the experimental material samples and minimize data errors between the experimental ore and the original material, the moderately weathered and fully weathered granite were mixed thoroughly using an XH-III three-dimensional mixer to obtain the comprehensive material sample for this experiment. Table 1 displays the chemical composition of weathered granite. It is evident that the $Al_2O_3$ content is 15.71% and the $Fe_2O_3$ content is 1.68%, both of which fail to meet the required standards for feldspar ceramic materials. Additionally, the $SiO_2$ content is 64.08%, which also falls below the minimum standards for quartz products. The results of the XRD analysis in Figure 1 indicate that the main mineral components of the original ore are quartz and potassium feldspar, with a quartz content of about 45%, a potassium feldspar content of about 25%, a mica content of about 15%, and a kaolinite content of 10%. There is also a small number of amorphous components, mainly due to the loss of mineral lattice or incomplete crystal cells during weathering, resulting in the presence of amorphous minerals. In the ceramic industry, the whiteness value of ceramic products determines their worth. The steps involved in detecting the whiteness of ceramic raw materials primarily include sample ball mill grinding, crushing, pressing forming, high temperature setting, and other processes [25,26]. Subsequently, the whiteness meter is utilized to measure the whiteness value, which is currently at 22.7%.

**Table 1.** Main chemical compositions of the weathered granite (wt. %).

| Element | $Al_2O_3$ | $SiO_2$ | $Fe_2O_3$ | $K_2O$ | $Na_2O$ | $TiO_2$ |
|---------|-----------|---------|-----------|--------|---------|---------|
| Content | 15.71 | 64.08 | 1.68 | 4.16 | 0.23 | 0.21 |

The granite samples reveal strong mineralogical and textural alterations as a result of intense physical and chemical weathering, which is expressed by a fine and uneven particle size distribution [27]. In order to further realize the particle size distribution characteristics of the weathered granite, a particle size screening test was conducted (Table 2). The coarse mineral particles with a diameter above 4.75 mm are relatively few, with the majority

distributed above 0.16 mm, accounting for 69.99% of the total content. At the same time, the content below 0.05 mm reaches 18.33%, indicating a high proportion of fine particles and a potentially large amount of slime in the original ore of the weathered granite. Impure iron is mainly concentrated in the fine particle size range, and alumina is also mainly concentrated in the fine particle size range. This pattern provides a basis for subsequent iron removal and the production of kaolin products.

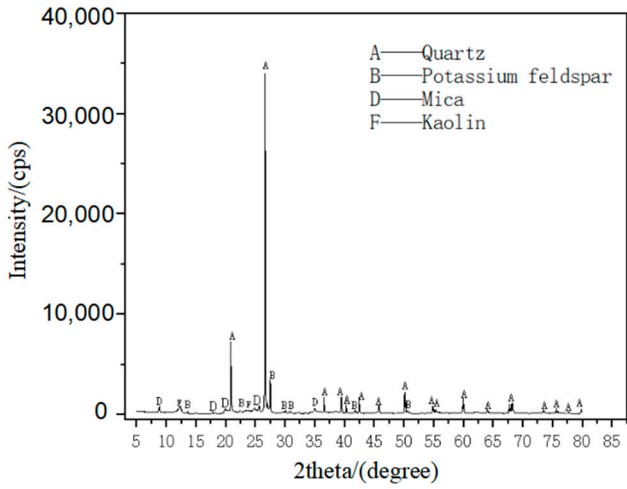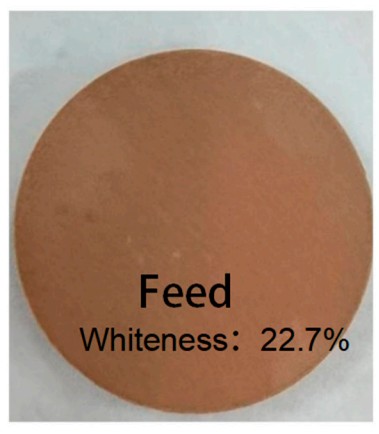

**Figure 1.** XRD and analysis of the weathered granite.

**Table 2.** Weathered granite grain size and major components: XRF analysis (wt. %).

| Particle Size Range/mm | $Fe_2O_3$ | $SiO_2$ | $Al_2O_3$ | $K_2O$ | Content |
|---|---|---|---|---|---|
| +4.75 | 1.74 | 93.98 | 20.70 | 2.94 | 3.16 |
| 4.75~2 | 1.47 | 85.83 | 9.67 | 2.50 | 27.83 |
| 2~1 | 1.44 | 82.50 | 10.49 | 4.90 | 13.86 |
| 1~0.42 | 1.93 | 74.98 | 14.43 | 7.76 | 16.35 |
| 0.42~0.16 | 2.39 | 68.07 | 19.07 | 9.55 | 11.95 |
| 0.16~0.074 | 3.22 | 59.64 | 25.76 | 10.26 | 5.89 |
| 0.074~0.05 | 3.78 | 56.50 | 29.92 | 8.58 | 2.13 |
| −0.05 | 4.25 | 51.33 | 37.52 | 5.17 | 18.33 |

The data in Table 3 show the results of the iron phase and silica phase analysis of the weathered granite. Among them, the predominant iron minerals in the rocks are primarily hematite and limonite, accompanied by a minor presence of iron silicate. Additionally, trace amounts of magnetite and metallic iron can be found, indicating that weathered granite is predominantly composed of weakly magnetic iron minerals. The ore exhibits a high total silica content of 70.19%, with free silica (quartz) comprising 42.88% and silicate-bound silica accounting for 27.31%.

**Table 3.** The distribution of iron and silicon phases in weathered granite (wt. %).

| Iron Phase Type | $Fe^{2+}$ (Hematite & Limonite) | mFe (Magnetic Iron) | MFe (Metal Iron) | Iron Silicate |
|---|---|---|---|---|
| Content | 88.98 | 0.425 | 0.425 | 10.17 |
| Silicon Phase Type | Free Silicon Dioxide (Quartz) | Silicon Dioxide in Silicate | Total Silicon Dioxide | |
| Silicon Dioxide Content | 42.88 | 27.31 | 70.19 | |

Concerning other physical and chemical properties of the weathered granite, the specific magnetization coefficient, measured using the LakeShore magnetometer, is $3.2 \times 10^{-5}$, and the Vickers hardness of the raw ore, measured using Innovatest Falcon507 equipment, is 783.94, indicating that even after weathering, the ore still has a high hardness, and the use of steel ball medium is considered for grinding.

The chemical reagents used in this experiment, such as dodecylamine (DDA), sodium dodecyl sulfonate (SDS), sodium oleate (NaOL), coconut oil amine (A CO), benzohydroxamic acid (BHA), sodium silicate, sulfuric acid ($H_2SO_4$), $Na_2CO_3$, NaOH, KCl, sodium hydrosulfite, and oxalic acid, were sourced from Shanghai Macklin Biochemical Technology Co., Ltd., Shanghai, China). All of the aforementioned agents were analytically pure. The tap water used in the experiment was obtained from Changsha city, Hunan province, where the laboratory is situated.

## 2.2. Bench-Scale Flotation Tests

The cone ball mill (XMQ, Wuhan Exploration Machinery Co., Ltd., Wuhan, China) was used for grinding the weathered granite. The hydrocyclone (CZ100, Changsha Mining and Metallurgy Research Institute Mining and Metallurgical Equipment Co., Ltd., Changsha, China) was used for desliming. Then, the ultra-fine high-gradient magnetic separator (DLSD, Yueyang Dali Shen Electromechanical Co., Ltd., Yueyang, China) was used for magnetic separation, followed by flotation. The single-slot flotation machine (XFD-IV) (Wuhan Exploration Machinery Co., Ltd., Wuhan, China) was used for Bench-scale flotation experiments, with a volume of 1 L, impeller speed of 1600 rpm, and slurry density of 28%.

Firstly, we used $Na_2CO_3$ as a pH regulator, KCl as a feldspar depressant, and a combination of NaOL/BHA/A CO as a collector to separate mica and other impurities containing iron silicate from quartz and feldspar. Then, we used diluted sulfuric acid as a pH regulator, sodium silicate as a quartz depressant, and DDA/SDS as a combination collector to float feldspar to achieve the separation of feldspar and quartz [28,29]. The concentrate and tailings were filtered, dried, and weighed for analysis.

## 2.3. Product Evaluation

The products of quartz, feldspar, and kaolin obtained through various tests were analyzed to judge whether they meet the marketing standards. In the following analysis, we used X-ray fluorescence spectrometers (Axios mAX, Dutch PANalytical Co., Ltd., Amsterdam, Dutch) and high-temperature box resistors (WEF.M25/16, Wofu Furnace Co., Ltd., Luoyang, China) to test the chemical composition of the conventional nine items. The whiteness of minerals was analyzed using a whiteness meter (SBDY-1, Shanghai Yuefeng Instrument Co., Ltd., Shanghai, China). The X-ray diffractometer (X'Pert3 POWder, Dutch PANalytical Co., Ltd., Amsterdam, Dutch) was utilized for both the mineral composition and phase analysis. The surface morphology and element distribution of mineral products were examined using scanning electron microscopy (JSM-IT500, Japanese JEOL, Tokyo, Japan), energy spectrometer (INCA X-ACT type), and electron probe (EPMA-1600 type).

## 3. Results and Discussion

### 3.1. Grinding

The grinding work index is a crucial indicator of ore grindability, and it provides essential data for accurately determining the mill diameter, medium diameter, and other key grinding parameters in concentrator design. This test was conducted to obtain the ball mill work index (WIB). And this was achieved by performing dry closed-circuit grinding with a ball mill, with the grinding cycle load reaching 250% [30,31]. The calculation formula for $W_{ib}$ is as follows:

$$W_{ib} = \frac{4.906}{p_1^{0.23} \cdot G_{bp}^{0.82} \cdot \left( \frac{1}{\sqrt{P_{80}}} - \frac{1}{\sqrt{F_{80}}} \right)} \tag{1}$$

The variables in the formula are defined as follows:

$W_{ib}$—Ball mill work index, kw·h/t;

$p_1$—Test sieve size, μm;

$G_{bp}$—The weight of the newly generated granular material beneath the test sieve with each rotation of the ball mill, g;

$P_{80}$—The particle size through which 80% of the material in the product passes, μm;

$F_{80}$—The particle size of 80% of the material in the mine, μm.

The test equipment used a φ 305 mm × 305 mm Bond power index ball mill. The diameter and quantity of steel balls added to the cylinder during the determination of the Bond power index are displayed in Table 4. The material sample was mixed and split in half. It was then dried at a low temperature and crushed to a size of −6 mesh (−3.35 mm). A sample of 1.0 kg was taken using the grid subdivision method, while the remaining crushed material sample was kept for future use. The particle size composition of the raw ore was determined through screening and analysis. The particle size distribution curve of the feed is shown in Figure 2a. Based on the grain size sieving curve of the raw ore, the $F_{80}$ value is 15.21 mesh, which is equivalent to 977.6 μm. The yield of the −200 mesh sample was 17.86%.

**Table 4.** The diameter and number of steel balls in the cylinder.

| Diameter (mm) | φ 36.5 | φ 30.2 | φ 25.4 | φ 19.1 | φ 16 | Total |
|---|---|---|---|---|---|---|
| Quantity (number) | 43 | 67 | 10 | 71 | 94 | 285 |

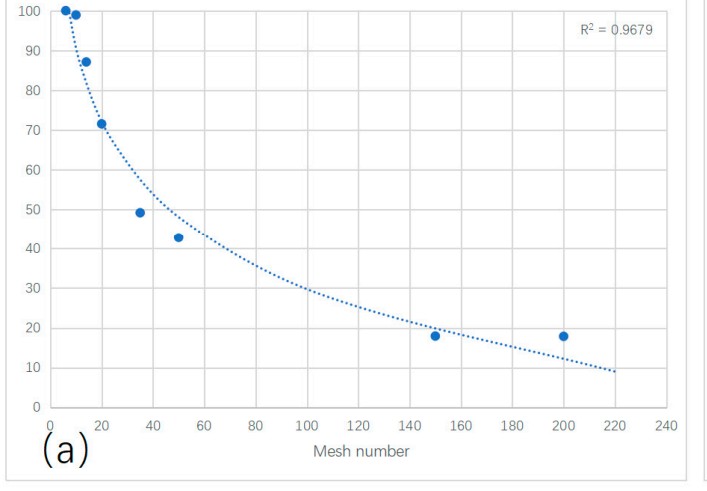
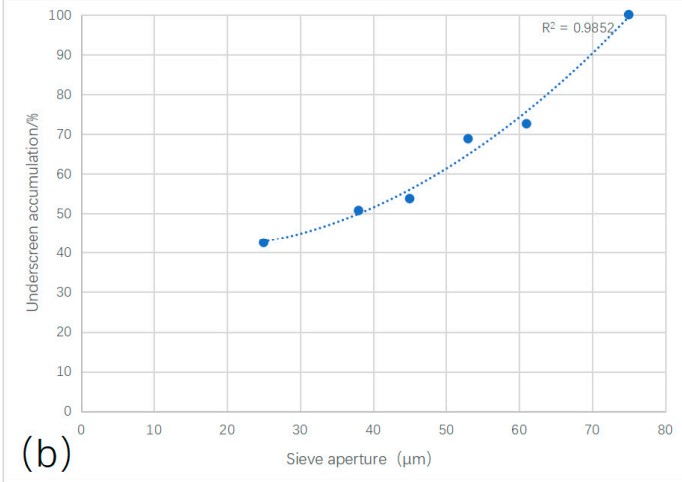

**Figure 2.** (**a**) Size curve of weathered granite sieve analysis; (**b**) particle size distribution curve after balance of product.

The 700 cm$^3$ material sample was weighed; the weight was 1024 g. The specific gravity of the sample was obtained: 1024/700 = 1.46 g/cm$^3$. The quality of the −200 mesh product in the feed was calculated as follows: sample weight × sample yield of −200 mesh under screen = 1024 × 17.86% = 182.87 g. During the measurement, a 700 cm$^3$ material sample must be kept in the ball mill, so that the expected product quantity can be calculated. According to the concept of the work index, the work index was obtained when the grinding reached 250% of the circulating load, and the expected product quantity was 1/3.5 of the quality of the 700 cm$^3$ sample. Therefore, the expected product quantity in this test was 1024/3.5 = 292.57 g. The number of cycles was set at five, and the test data were measured according to the existing steps (Table 5).

**Table 5.** Data of 75µm Bond ball milling power index test of weathered granite.

| Cycle Order | Number of Revolutions (r/min) | $M_f$ (g) | $M_p$ (g) | $M_O$ (g) | $G_{bp}$ (g/r) | Circulating Load (%) |
|---|---|---|---|---|---|---|
| 1 | 200 | 182.87 | 396.57 | 213.70 | 1.0685 | 158.21 |
| 2 | 205 | 72.88 | 338.35 | 265.47 | 1.2950 | 202.65 |
| 3 | 178 | 61.81 | 309.34 | 247.53 | 1.3906 | 231.03 |
| 4 | 169 | 56.57 | 292.4 | 235.83 | 1.3954 | 250.21 |
| 5 | 170 | 53.14 | 293.2 | 240.06 | 1.4121 | 249.25 |

The variables in Table 5 are defined as follows:

$M_f$—Material weight under −200 mesh sieve, g;

$M_P$—Mill product weight −200 mesh under screen material, g;

$M_O$—Net grinding production −200 mesh material weight under screen, g;

$G_{bp}$——200 mesh underscreen material weight generated per turn, g.

The balance criterion is judged according to the allowable error of the cycle load and $G_{bp}$ value in the last two cycles. As shown in Table 5, the load for the fourth and fifth cycles is 250.21% and 249.25%, respectively. The average cycle load is 249.73%. The values for $G_{bp}$ are 1.3954 and 1.4121, respectively, with an average $G_{bp}$ of (1.4121 + 1.3954)/2 = 1.40375. Therefore, the error of the $G_{bp}$ value is as follows:

$$(\text{maximum} - \text{minimum})/\text{average};$$
$$= (1.4121 - 1.3954)/1.40375 * 100\%;$$
$$= 1.189\% \text{ (This value is less than the allowable error of } G_{bp} \text{ by 3 \%)}.$$

The products are mixed evenly under the 4th and 5th -200 mesh sieves. Then, a 200 g sample is taken using the binomial method for sieving. The particle size curve of the sieved product after reaching equilibrium is shown in Figure 2b. From the curve, $P_{80}$ can be determined to be 63.92 µm.

From Equation (1), the Bond ball grinding index $W_{ib}$ = 14.79 kWh/t.

A high $W_{ib}$ value indicates that the weathered granite has a high hardness and is a difficult rock to grind. However, the weathered granite contains a high amount of primary slime. Therefore, the process involves initially using high-frequency vibration to screen and classify the ore, followed by grinding to prevent excessive grinding. This method also allows for the selective separation of weathered fine-grained minerals, which is advantageous for the subsequent preparation of coarse-grained mineral iron removal and kaolin products. To determine the appropriate grinding fineness, a grinding condition test is conducted. Five samples, each weighing 500 g and with the primary slime removed, are subjected to steel ball milling for different durations: 3, 5, 7, 9, and 11 min. The resulting secondary slime, with a particle size below 0.037 mm, is obtained through re-desliming. The grind materials are then analyzed, and the results are presented in Table 6. With the prolongation of grinding time, the number of fine particles increased significantly, and the content of secondary slime increased significantly due to the dissociation of iron-bearing minerals during grinding. But the total iron levels also show signs of rising. This may be caused by the steel balls rubbing against each other during the grinding process to produce metallic iron. Taking into account the cost and efficiency of actual production and application in the mine, a grinding time of 7 min is selected as the optimal choice.

**Table 6.** Grinding time and test results.

| Grinding Time/min | −0.074 mm Content/% | Secondary Mine Slime Content/% | $TFe_2O_3$/% |
|---|---|---|---|
| 3 | 57.33 | 10.23 | 0.56 |
| 5 | 63.21 | 12.34 | 0.73 |
| 7 | 67.23 | 14.02 | 0.58 |
| 9 | 69.29 | 20.24 | 0.75 |
| 11 | 73.98 | 24.23 | 0.68 |

### 3.2. Desliming

After the grinding process, a significant amount of slime was produced from weathered granite, which necessitates an efficient desludging procedure [32]. The CZ100 model high-efficiency hydrocyclone was utilized, featuring a cone angle of 10 degrees and a column length of 150 mm. This hydrocyclone was primarily employed for desludging and concentrating fine-grained materials. By adjusting the diameter of the sand outlet of the hydrocyclone, various desludging effects could be achieved. The overflow pipe diameter is 22 mm, cone angle is 10 degrees, and the feed pressure is 0.2 MPa. Five different conditions were set for the sand outlet diameter, namely 12 mm, 14 mm, 16 mm, 18 mm, and 20 mm, and desludging experiments were conducted on weathered granite. The overflow and sand products were obtained and analyzed using a BT-9300ST laser particle size analyzer, and the product yield and XRF results were shown in Table 7.

**Table 7.** Desliming conditions test product yield table.

| Serial Number | Parameter | Product | Concentration/% | Productivity/% | $TFe_2O_3$/% | $Al_2O_3$/% |
|---|---|---|---|---|---|---|
| 1 | $d_0$ = 22 mm $d_s$ = 12 mm P = 0.2 MPa | Overflow 1 | 2.92 | 15.79 | 5.42 | 0.21 |
| | | Riffling 1 | 46.78 | 84.21 | 1.74 | 0.19 |
| | | Feed 1 | 13.87 | 100.00 | 2.42 | 14.20 |
| 2 | $d_0$ = 22 mm $d_s$ = 14 mm P = 0.2 MPa | Overflow 2 | 3.08 | 14.30 | 5.39 | 0.18 |
| | | Riffling 2 | 35.92 | 85.70 | 1.80 | 0.22 |
| | | Feed 2 | 14.22 | 100.00 | 2.42 | 14.20 |
| 3 | $d_0$ = 22 mm $d_s$ = 16 mm P = 0.2 MPa | Overflow 3 | 3.04 | 13.28 | 5.82 | 0.21 |
| | | Riffling 3 | 32.50 | 86.72 | 1.64 | 0.23 |
| | | Feed 3 | 14.20 | 100.00 | 2.42 | 14.20 |
| 4 | $d_0$ = 22 mm $d_s$ = 18 mm P = 0.2 MPa | Overflow 4 | 2.81 | 10.40 | 5.88 | 0.22 |
| | | Riffling 4 | 27.35 | 89.60 | 1.89 | 0.20 |
| | | Feed 4 | 14.33 | 100.00 | 2.42 | 14.20 |
| 5 | $d_0$ = 22 mm $d_s$ = 20 mm P = 0.2 MPa | Overflow 5 | 2.85 | 9.67 | 5.85 | 0.23 |
| | | Riffling 5 | 25.42 | 90.33 | 2.26 | 0.19 |
| | | Feed 5 | 14.39 | 100.00 | 2.42 | 14.20 |

The total size of the feed ore is fine, and the $Fe_2O_3$ content is 2.42%, making it suitable for hydraulic cyclone classification desliming. By comparing the data from Figure 3 the five groups of tests, it can be concluded that the yield of the overflow product, or slime, decreases as the diameter of the sedimentation port increases. Additionally, the particle size of the sedimentation product also increases with the increase in the diameter of the sedimentation port. However, the $Fe_2O_3$ content of the sedimentation product does not show a clear trend. It can be inferred that a larger diameter sedimentation port is not conducive to effective classification. The desilting yield at 12 mm is 15.79%. Taking into

account the product yield and the $Fe_2O_3$ and $Al_2O_3$ content, the best desliming process condition is selected as a settling mouth diameter of 12 mm.

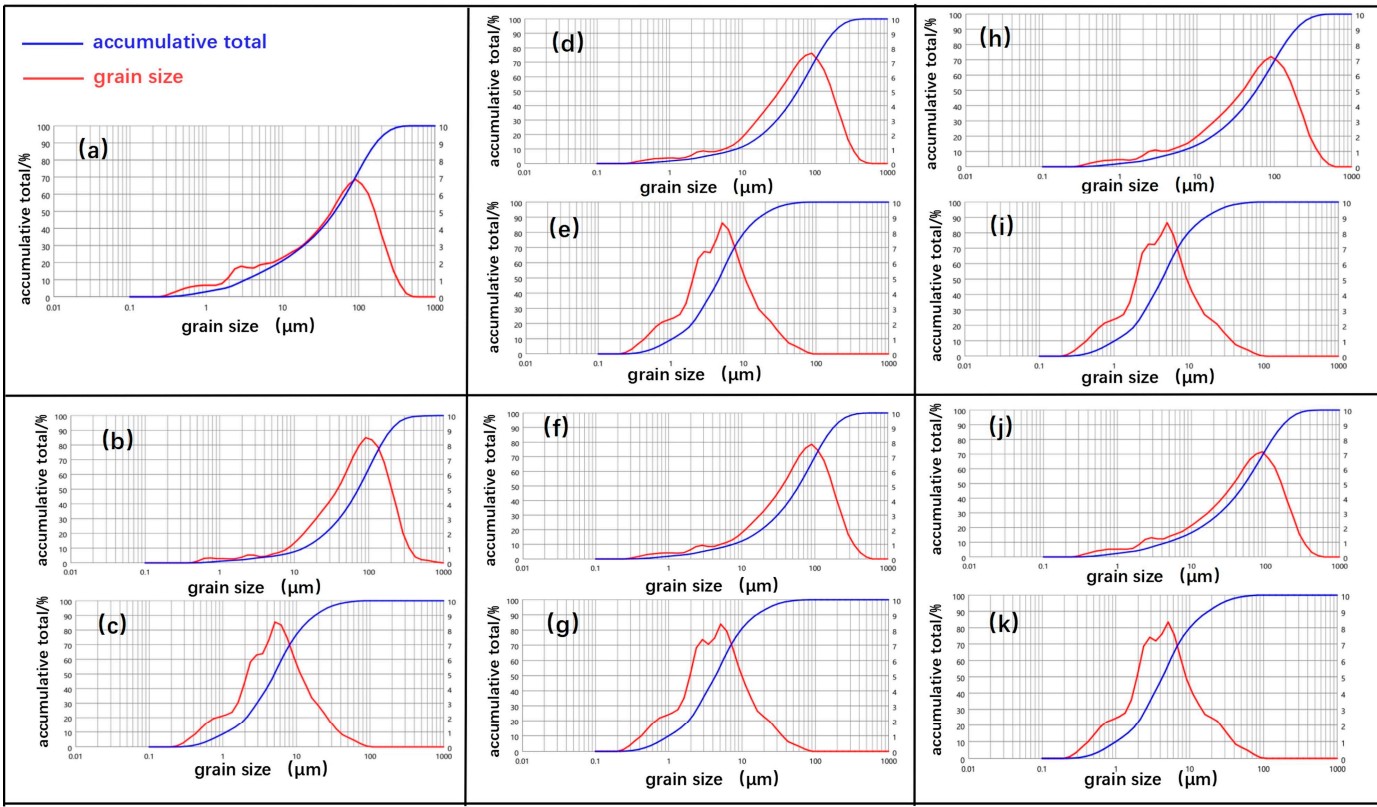

**Figure 3.** Particle size distribution of hydrocyclone products: (**a**) feed product, (**b**) overflow 1 product, (**c**) riffling 1 product, (**d**) overflow 2 product, (**e**) riffling 2 product, (**f**) overflow 3 product, (**g**) riffling 3 product, (**h**) overflow 4 product, (**i**) riffling 4 product, (**j**) overflow 5 product, and (**k**) riffling 5 product.

Quartz and feldspar products require low iron content, so it is necessary to completely remove iron impurities, most of which are weakly magnetic ores. It is found that the high gradient wet magnetic separator with a magnetic induction intensity of 1.5 T (Tesla) is more excellent for strong magnetic separation. The weathered granite has a small particle size and high iron content. The effect of the magnetic separation is limited, and the combination of magnetic separation and flotation can improve the efficiency of beneficiation.

### 3.3. Flotation Test

Based on the existing studies, the mixed collectors are utilized for the purpose of separating iron-bearing minerals like mica and hematite from quartz and feldspar through flotation on a 1 L flotation machine. In this particular process, the collector is pre-arranged and subsequently added to the pulp, as depicted in Figure 4. The study examines the impact of various factors, including pulp pH, NaOL/BHA/A CO mixture ratio, and the quantity of the collector. The grade and recovery rate of iron (Fe) in the tailings are employed as indicators to evaluate the flotation process.

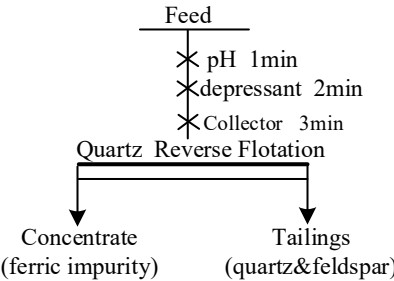

**Figure 4.** Quartz reverse flotation test flow diagram.

Figure 5a illustrates the impact of pulp pH on the recovery and grade of $Fe_2O_3$ in tailings when using a mixed collector of NaOL/BHA/A CO. The other parameters in the flotation process remain constant: the depressant KCl and sodium silicate are used at 1000 g/t and 300 g/t, respectively, the collector NaOL is used at 100 g/t, BHA at 40 g/t, and A CO at 1 g/t. As depicted in Figure 5a, within the pH range of 6 to 12, the recovery rate and grade of $Fe_2O_3$ initially decrease and then increase with increasing pulp pH. The removal of iron-bearing minerals is particularly effective at a pH of 8 to 9, and the iron grade is at its lowest when the pH is 8.3. Therefore, the optimal pH for quartz reverse flotation to remove iron is 8.3.

Figure 5b shows the impact of different mixtures of NaOL/BHA/A CO on the recovery and grade of $Fe_2O_3$ in tailings at a pulp pH of 8.3. The fixed amount of the mixed collectors is 200 g/t, and other conditions in the flotation process remain constant. It is evident that as the proportion of sodium oleate increases, the recovery rate and grade of $Fe_2O_3$ decrease significantly. The most effective removal of iron-bearing minerals occurs when the mixing ratio of NaOL: BHA: A CO is 3:1:0.05, and further increases in the proportion of each reagent worsen the removal effect. Coconut oil amine (A CO), a primary amine compound commonly used as an emulsifier, can cause pulp flocculation if overdosed, thereby affecting flotation effectiveness [33]. Additionally, benzohydroxamic acid is expensive, so the optimal ratio of the mixed collector is NaOL: BHA: A CO = 3:1:0.05.

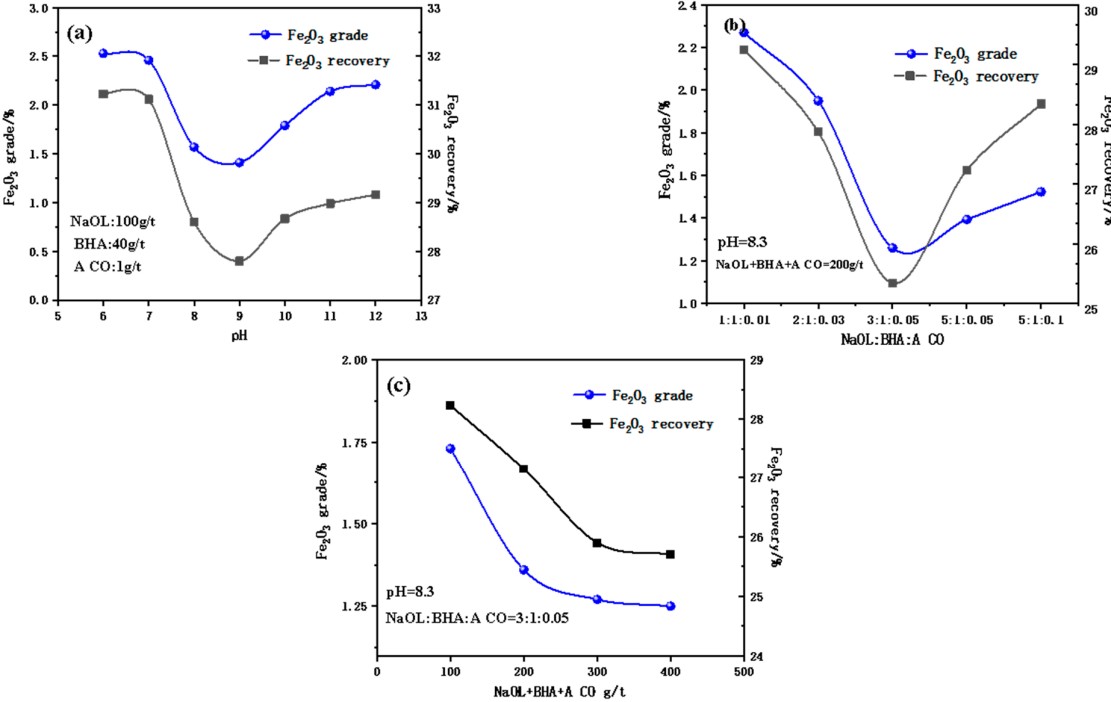

**Figure 5.** The recovery rate and grade of $Fe_2O_3$ varied depending on (**a**) the pulp pH, (**b**) the NaOL/BHA/A CO ratio, and (**c**) the amount of collector.

When the pH of the pulp is 8.3 and the mixing ratio of NaOL/BHA/A CO is 3:1:0.05, the influence of the collector amount on the recovery and grade of $Fe_2O_3$ in the tailings is depicted in Figure 5c. The other conditions in the flotation process remain constant, as mentioned above. It can be observed that as the collector amount increases within the range of 100–400 g/t, the recovery rate and grade of $Fe_2O_3$ continue to decrease. The removal effect of iron-bearing minerals reaches a balance at 300 g/t collector amount, and the effect of iron removal fluctuates minimally with further increases in the collector amount. Therefore, the optimal number of mixed collectors is determined to be 300 g/t. In summary, the best flotation reagent process is when the pH is 8.3 and the dosage of mixed collector (NaOL: BHA: A CO = 3:1:0.05) is 300 g/t.

The aforementioned tailings were subjected to further flotation. The mixed collector was used to separate feldspar from quartz. Sodium silicate is used as a depressant of quartz. During this process, the collecting agents were added to the ore pulp in the sequence of DDA and SDS, as depicted in Figure 6. The effects of various factors, such as ore pulp pH, DDA/SDS mixing ratio, and collecting agent dosage, were examined, and the flotation process was assessed based on the grade and recovery rate of the quartz product.

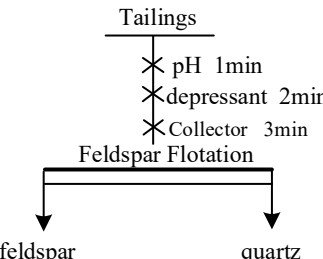

**Figure 6.** Feldspar flotation test flow diagram.

The effect of pulp pH on the recovery and grade of $SiO_2$ in quartz products using a mixed collector of DDA/SDS was investigated. The other parameters in the flotation process remained unchanged: the depressant sodium silicate was used at a quantity of 300 g/t, the collector DDA was used at a quantity of 60 g/t, and the SDS was used at a quantity of 40 g/t. As shown in Figure 7a, within the pH range of 2 to 6, the recovery of $SiO_2$ initially decreases and then increases with an increase in pulp pH. Similarly, the grade of $SiO_2$ initially increases and then decreases. Notably, at pH levels of 2 to 3, there is a significant enrichment effect on silica, and the highest grade of $SiO_2$ is achieved at a pH of 2.7. Therefore, the optimal pH for the flotation separation of quartz and feldspar is 2.7.

Figure 7b illustrates the impact of $SiO_2$ recovery and grade on collector quartz products made up of different mixing ratios of DDA/SDS when the pulp pH is 2.7. The fixed amount of the mixed collectors is 100 g/t, and the other conditions in the flotation process remain constant. It can be observed that the enrichment effect of quartz is more pronounced when the DDA content is high, indicating a better collection efficiency of DDA and stronger selectivity of SDS. The highest $SiO_2$ grade is achieved when the DDA:SDS ratio is 3:1. Therefore, the optimal ratio for the mixed collectors is DDA: SDS = 3:1.

When the mixed ratio of the mixed collectors DDA/SDS is 3:1 and the pH of the pulp is 2.7, the influence of the collector amount on the recovery and grade of $SiO_2$ in quartz products is studied, as shown in Figure 7c. Other conditions in the flotation process are kept constant, as mentioned above. It can be observed that within the range of 50 to 300 g/t collector dosage, the recovery rate of $SiO_2$ continues to decrease as the collector dosage increases. Initially, $SiO_2$ increases, but the curve stabilizes when the dosage reaches 200 g/t. Taking into account that DDA itself has strong foaming properties, an excessive dosage will affect the flotation effect [28]. Therefore, the optimal dosage of the mixed collectors is 200 g/t.

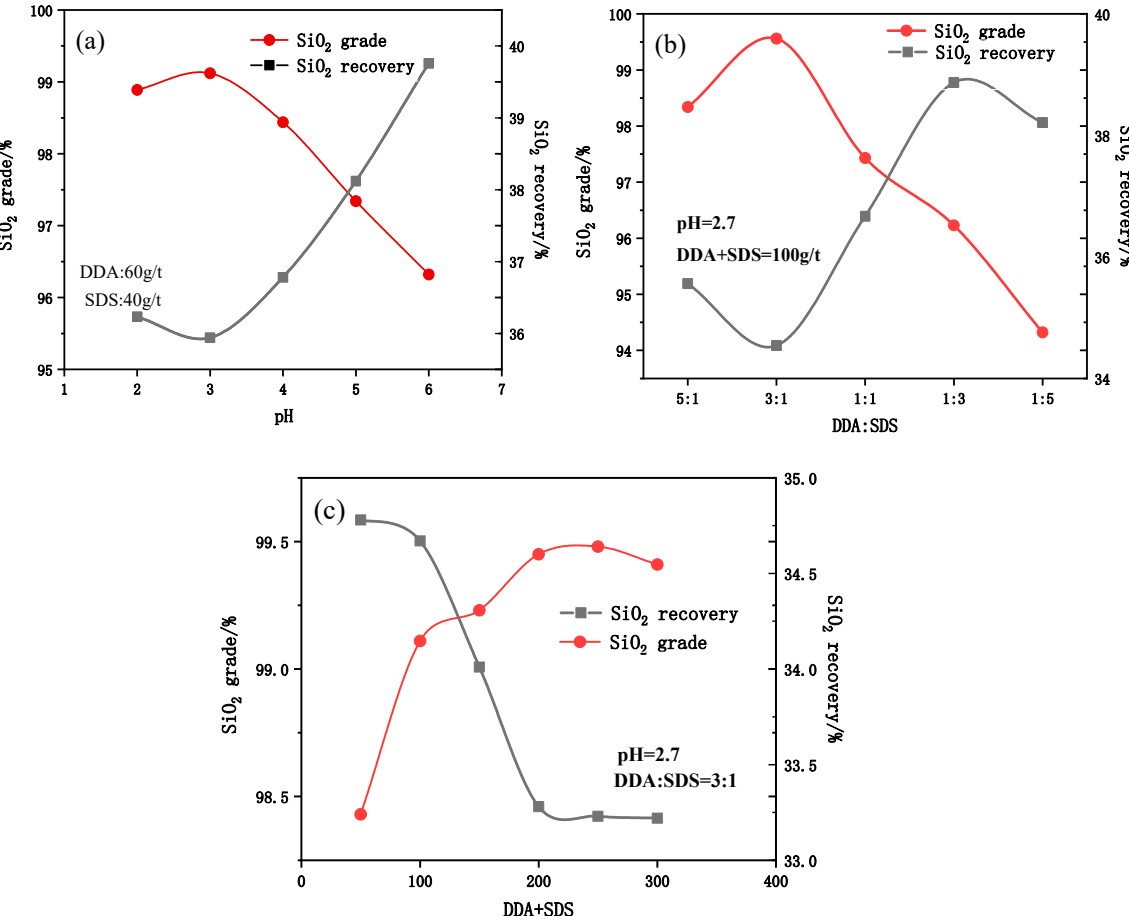

**Figure 7.** The recovery rate and grade of SiO$_2$ varied depending on (**a**) the pulp pH, (**b**) the DDA/SDS ratio, and (**c**) the amount of collector.

### 3.4. Comprehensive Recycling Process

On the basis of the above research, further research on the test flow was carried out to obtain an efficient selection process scheme. Figure 8 illustrates the mass balance chart for the comprehensive utilization of weathered granite.

For the fine grade raw ore obtained using the high-frequency vibration fine screen, the magnetic wet separation was carried out twice by using a high gradient magnetic separator, and the magnetic field intensity was controlled at 1.5 T (Tesla). Kaolin products are rich in kaolinite, and kaolinite was mainly concentrated in the fine grade raw ore, so fine grade raw ore was used to produce kaolin products. The product standard of kaolin requires a particle size of less than 800 mesh, while meeting the low iron content. Therefore, the non-magnetic products needed to be re-ground and modified with 5% sodium bisulfite and 5% oxalic acid to obtain qualified kaolin products.

The rough concentrate of quartz obtained using reverse flotation contained a large amount of iron minerals. It was mixed with magnetic products to form a mixture of minerals including mica, low-grade feldspar, quartz, and iron impurities, which was used as raw material for aerated bricks, effectively achieving zero-waste production. The feldspar and quartz obtained from four flotation cycles, and with the fine-grade slurry removed by a hydrocyclone, are mixed together. This product also had a high iron content and was subjected to magnetic separation using a high-gradient magnetic separator twice, with a magnetic field intensity of 1.5 T. The resulting magnetic product was also used as raw material for aerated bricks. The non-magnetic product was then subjected to a rough four-stage reverse flotation process to remove iron, resulting in feldspar products and by-products, which were sold as ceramic raw materials of different grades.

This production process produced a variety of products including quartz, feldspar, kaolin, aerated brick raw materials, and feldspar by-products. It maximized the recovery of valuable components and improved resource utilization efficiency. Next, each product would undergo a qualification analysis.

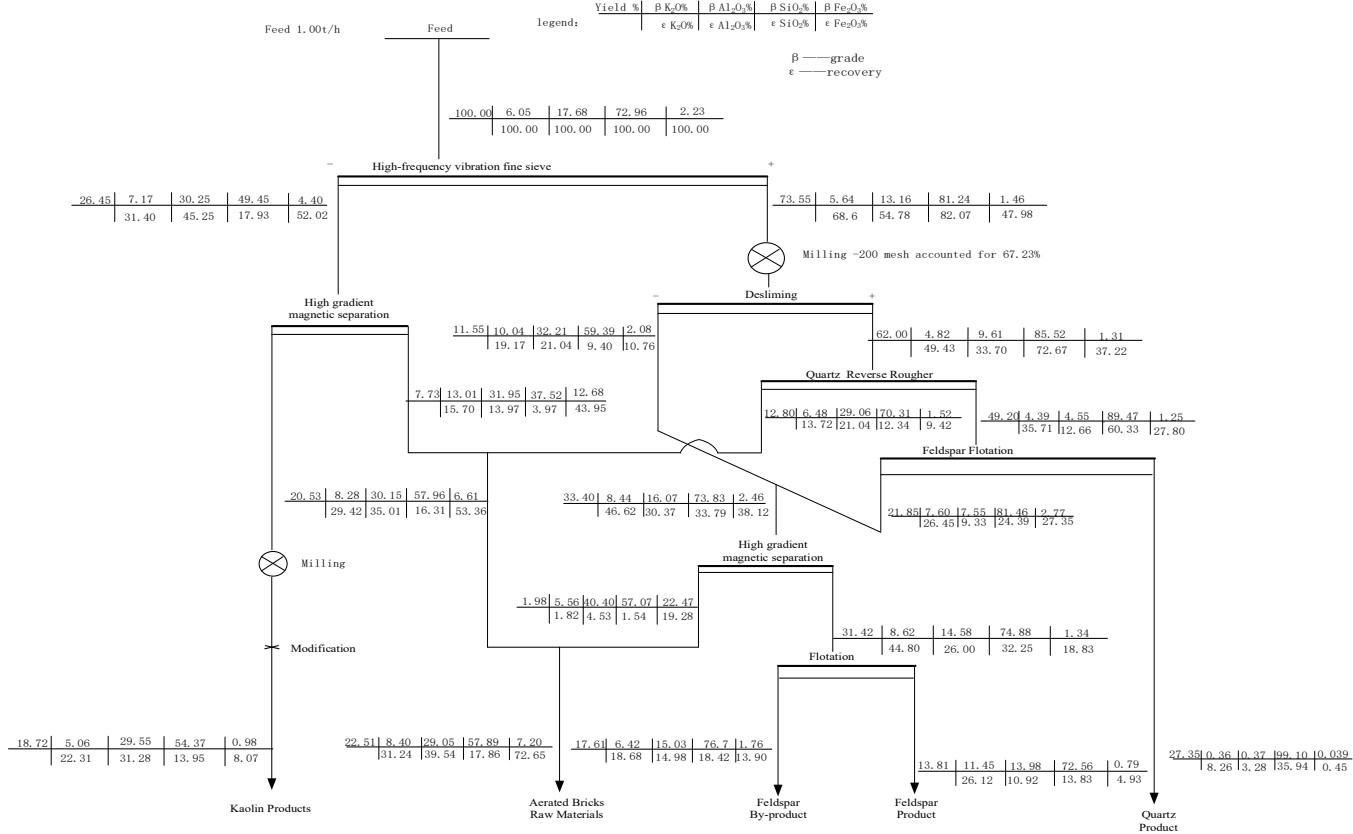

**Figure 8.** Mass balance chart for comprehensive utilization of weathered granite.

*3.5. Product Analysis*

The nine routine chemical components are important criteria for testing the quality of non-metallic raw materials in China. They include the content of LOI (1025 °C), $Al_2O_3$, $SiO_2$, $Fe_2O_3$, CaO, MgO, $K_2O$, $Na_2O$, and $TiO_2$ [34]. The analysis results of the five products are shown in Tables 8 and 9. Among them, the silica content of quartz products is more than 99%, and the whiteness is 95.5, which meets the standard of refined quartz sand. Feldspar products have an iron content of less than 0.5% and a whiteness of 35.9, which is suitable for the production of glazes and flat glass as high-quality feldspar products. Feldspar by-products with an iron content of less than 1% can be used as tertiary feldspar products for enamel production. The raw materials of aerated brick have a high content of silicon and iron, which meet the preparation standards of aerated brick. Kaolin has a high content of aluminum and silicon, and its whiteness is 61.9. In order to determine whether it meets the product standards of kaolin, a particle morphology analysis is required.

**Table 8.** Product chemical composition routine nine items (wt. %).

| Product | Loss on Ignition (LOI, 1025 °C) | $Al_2O_3$ | $SiO_2$ | $Fe_2O_3$ | CaO | MgO | $K_2O$ | $Na_2O$ | $TiO_2$ |
|---|---|---|---|---|---|---|---|---|---|
| quartz | 0.22 | 0.23 | 99.51 | 0.018 | 0.01 | 0.01 | 0.11 | <0.01 | <0.01 |
| feldspar | 1.49 | 12.42 | 76.07 | 0.46 | 0.11 | 0.04 | 8.63 | 0.36 | 0.03 |
| kaolin | 9.92 | 32.54 | 51.48 | 2.21 | 0.14 | 0.19 | 4.83 | 0.15 | 0.11 |
| feldspar by-products | 1.67 | 18.74 | 76.31 | 0.98 | 0.23 | 0.05 | 6.93 | 0.54 | 0.12 |
| aerated brick raw materials | 2.32 | 29.05 | 57.89 | 7.2 | 1.61 | 0.34 | 8.4 | 0.98 | 0.09 |

**Table 9.** Product whiteness analysis results.

| Product | Whiteness |
|---|---|
| quartz | 95.5 |
| feldspar | 35.9 |
| kaolin | 61.9 |
| feldspar by-products | 28.3 |
| aerated brick raw materials | 2.32 |

Scanning electron microscopy was utilized to observe the morphology of kaolin products, as depicted in Figure 9. No iron impurities were detected in the kaolin products, but there were trace amounts of feldspar carbonate minerals with extremely fine particle sizes. These minerals were found to be evenly distributed within the kaolinite, resulting in a high overall kaolinite content. The kaolin particles exhibited a fine plate structure, indicating a high level of product purity that meets the standards for qualified kaolin.

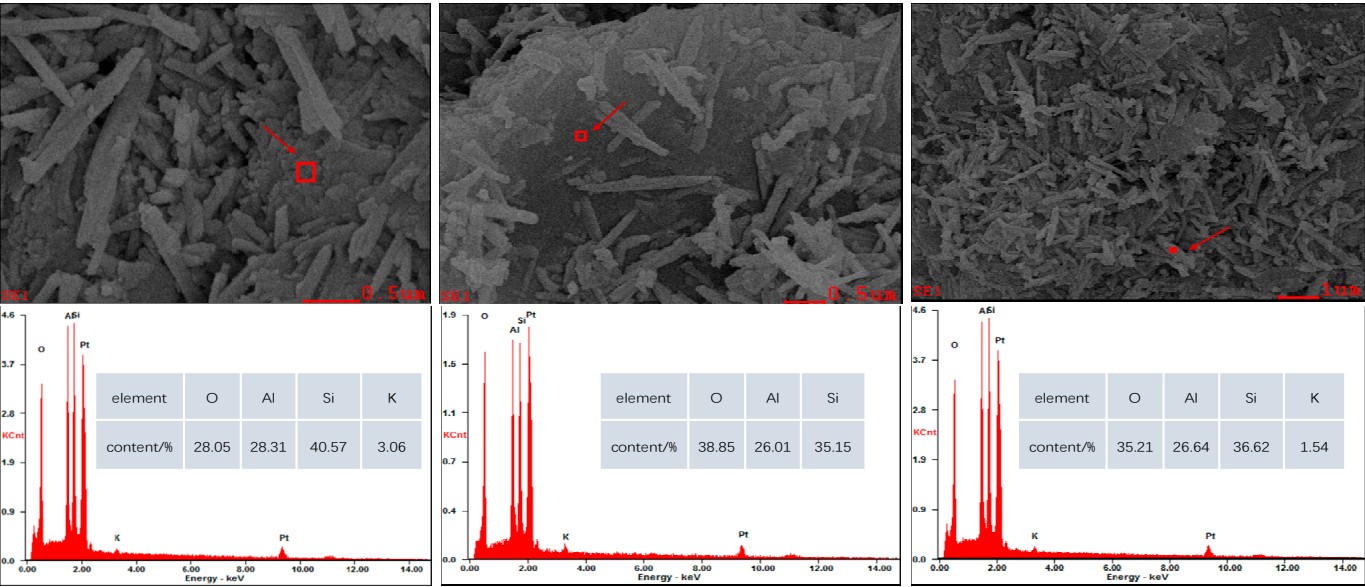

**Figure 9.** Scanning electron microscope and energy spectrum analysis of kaolin products.

## 4. Conclusions

(1) This study focused on weathered granite as the research object and utilized it as a resource. Granite possesses characteristics such as a hard and dense texture, high strength, corrosion resistance, and wear resistance. The study identified high value-added products that can be utilized in industries such as building materials, composite materials, and fine ceramics. The various minerals contained in the weathered granite can be effectively recovered through a composite procedure involving sifting, grinding, desliming, magnetic separation, and acid-based mixed flotation.

(2) The grinding test revealed that the grinding work index of weathered granite is 14.79 kWh/t, and the optimal grinding time is 7 min. The desliming test indicated that the ideal desliming process condition is to select a settling mouth diameter of 12 mm for the hydrocyclone. The flotation tests demonstrated that the best process conditions for the flotation of iron-bearing minerals, such as mica, from quartz and feldspar are a pH of 8.3, a mixing ratio of NaOL/BHA/A CO of 3:1:0.05, and a dosage of 300 g/t. The optimal process conditions for flotation separation of quartz and feldspar are a pH of 2.7, a DDA/SDS mixing ratio of 3:1, and a dosage of 200 g/t.

(3) The process plan achieves the objective of efficiently recovering feldspar, quartz, and kaolin from weathered granite. It also ingeniously considers the intermediate value of mineral components such as feldspar by-products and raw materials for aerated bricks. This

is a clean and environmentally friendly process plan that makes full use of all components without any solid waste accumulation. It is expected to bring hundreds of millions of revenue after it is put into production.

**Author Contributions:** H.H. conceived of and designed the experiments; S.L. performed the experiments and wrote the paper; H.G. contributed materials; N.Z. and L.L. modified the paper. All authors have read and agreed to the published version of the manuscript.

**Funding:** This research received no external funding.

**Data Availability Statement:** Data sharing is not applicable to this article.

**Acknowledgments:** We are grateful to the Engineering Research Center of the Ministry of Education for Carbon Emission Reduction in Metal Resource Exploitation and Utilization of Central South University for technical support.

**Conflicts of Interest:** The authors declare no conflicts of interest.

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
