# Peer review of "Efficient Recovery of Feldspar, Quartz, and Kaolin from Weathered Granite"

_minerals, doi:10.3390/min14030300_

Round 1

Reviewer 1 Report

Comments and Suggestions for Authors

My suggestions to the authors are given in the attached file.

Comments on the Quality of English Language

English of the paper is very difficult to understand/incomprehensible and needs extensive revision as shown in the attached file.

Author Response

Dear reviewer,

We feel great thanks for your professional review work on our article.The pdf file you marked has been of great help to the revision of our paper. As you are concerned, there are several problems that need to be addressed. According to your nice suggestions, we have made extensive corrections to our previous draft. Please see the attachment.

Thank you again for your careful and patient review of our manuscript.

Best regards.

Yours sincerely,

Hongjun Huang, Shihan Li, Haoran Gou, Ning Zhang and Liming Liu

Address:Central South University, Changsha, China

E-mail: [email protected]; [email protected]

Reviewer 2 Report

Comments and Suggestions for Authors

The manuscript documents a technically well-supported study with significant impact on the beneficiation and recovery of industrial minerals, despite reduced scientific innovation. The potential economic impact of the results obtained is evident, as well as the promotion of eco-efficient practices in the use of natural materials usually classified as waste.

The manuscript is well structured and, in general, well written. The objectives are clear, as is the methodology used, allowing all stages of the procedure to be followed in great detail and the results obtained in each of them assessed. The study's conclusions are therefore well anchored in the reported results.

Suggestions:

1) The term "ore" (Lines 70, 72, 78, 175, 178, 181, 298, 336, 339) should be avoided and replaced with "material"

2) The statement "Weathered granite undergoes strong physical and chemical weathering" (line 90) is redundant. I suppose the authors mean to say: "The granite samples reveal strong mineralogical and textural alteration as a result of intense physical and chemical weathering, which is expressed by fine and uneven particle size distribution".

3) Line 98, "whereas" instead of "while"

4) Line 101, "Table" instead of "TTable"

5) Line 102, "Table 3 display the" instead of "The data in Table 3 show the analysis results of the"

6) Section 2.2 (lines 119-133). In my opinion, all verbs must be conjugated in the past, as it is a report that was made.

7) Section 2.3 (lines 134-146). In my opinion, all verbs must be conjugated in the past, as it is a report that was made. The first sentence needs a small wording adjustment; for example: "Products obtained from various experiments were analyzed and tested, investigating whether the three 135 products of quartz, feldspar, and kaolin meet the market trading standards.

8) Lines 162-164. In my opinion, all verbs must be conjugated in the past, as it is a report that was made.

9) Lines 178-179. In my opinion, all verbs must be conjugated in the past, as it is a report that was made.

10) Lines 182-182. The sentence must be rewritten.

11) Lines 197-203. In my opinion, all verbs must be conjugated in the past, as it is a report that was made. Some of these sentences lack writing polish.

12) Section 3.2 (lines 219-230). In my opinion, all verbs must be conjugated in the past, as it is a report that was made.

13) Line 241: "Considering" instead of "Taking into account"

14) Section 3.3 (lines 253-260). In my opinion, all verbs must be conjugated in the past, as it is a report that was made.

15) Section 3.4 (lines 332-250). In my opinion, all verbs must be conjugated in the past, as it is a report that was made.

16: Line 383. "material" instead of "object"

17: Line 384. "shows" instead of "possesses"

18: Line 400. "responsible" instead of "clean"

Comments on the Quality of English Language

It will be necessary to carefully review the formatting and writing errors (namely spaces between words, commas, and other minor aspects, but which always cause problems with the fluidity of reading). For suggestions, please see the above section.

Author Response

Dear reviewer,

We feel great thanks for your professional review work on our article. As you are concerned, there are several problems that need to be addressed. According to your nice suggestions, we have made extensive corrections to our previous draft. Please see the attachment.

Thank you again for your careful and patient review of our manuscript.

Best regards.

Yours sincerely,

Hongjun Huang, Shihan Li, Haoran Gou, Ning Zhang and Liming Liu

Address:Central South University, Changsha, China

E-mail: [email protected]; [email protected]

Reviewer 3 Report

Comments and Suggestions for Authors

It is suggested that the economic viability of recycling weathered granite into useful minerals be discussed in the final section of this manuscript. 

Comments on the Quality of English Language

The language expression should be checked and polished throughout the manuscript.

Author Response

Dear reviewer,

We feel great thanks for your professional review work on our article.As you are concerned, there are several problems that need to be addressed. According to your nice suggestions, we have made extensive corrections to our previous draft. Please see the attachment.

Thank you again for your careful and patient review of our manuscript.

Best regards.

Yours sincerely,

Hongjun Huang, Shihan Li, Haoran Gou, Ning Zhang and Liming Liu

Address:Central South University, Changsha, China

E-mail: [email protected]; [email protected]

Reviewer 4 Report

Comments and Suggestions for Authors

Manuscript ID: minerals-2901341

Title: Efficient Recovery of Feldspar, Quartz, and Kaolin from Weathered Granite

Authors: Hongjun Huang et al.

Line 30, 41. Avoid more than 3 references to a fact in one sentence. A maximum of 3 in a sentence is allowed for Minerals. Describe this information in detail.

The introduction contains general words. There is no detailed description of the enrichment methods with indication of technological parameters. It is recommended that the authors add numerical values to the text of this section.

What is the detail novelty of this research? Add this information.

Table 1. What is the LOI content?

Section 3.2. Can authors add a figure or blueprint of CZ100 model hydrocyclone?

Figure 7. May be authors used the one figure or 5 figures with particles size distribution data? Now very difficult to compare the results. Maybe 1 product: Overflow and Riffling in one figure.

Section 3.3. Authors should add information about yield of the Fe2O3 and SiO2 products.

Table 8. What is about concentrate (ferric impurity) chemical composition (wt.%)?

What is the phase from of Fe in the concentrate (ferric impurity)? Add XRD patterns and SEM image of this product.

Conclusions are too general, add paragraphs 1) 2) 3) with detailed results of the study using numerical values obtained after enrichment of granite

Technical errors:

Line 30-31. Write the full chemical formulas of feldspar, quartz, mica, and kaolinite in brackets.

Line 211, 216, etc. Authors should use “min”, not “minutes”.

Line 114. Use subscripts for chemical formulae.

References didn’t write in the Minerals style, please improve this section.

Author Response

(The authors gave the same response as above.)

Round 2

Reviewer 1 Report

Comments and Suggestions for Authors

I accept all the corrections made by the authors after my previous suggestions.

Comments on the Quality of English Language

A final check on the quality of the English language is necessary.

Reviewer 4 Report

Comments and Suggestions for Authors

The authors answered all questions in detail and greatly improved the article quality. Now this article can be accepted.